# Using Deep Learning to Detect Defects in Manufacturing: A Comprehensive Survey and Current Challenges

**DOI:** 10.3390/ma13245755

**Published:** 2020-12-16

**Authors:** Jing Yang, Shaobo Li, Zheng Wang, Hao Dong, Jun Wang, Shihao Tang

**Affiliations:** 1School of Mechanical Engineering, Guizhou University, Guiyang 550025, China; jyang23@gzu.edu.cn (J.Y.); gs.wangz17@gzu.edu.cn (Z.W.); gs.hdong19@gzu.edu.cn (H.D.); gs.wangjun19@gzu.edu.cn (J.W.); 2Guizhou Provincial Key Laboratory of Public Big Data, Guizhou University, Guiyang 550025, China; 3Key Laboratory of Advanced Manufacturing Technology of Ministry of Education, Guizhou University, Guiyang 550025, China; shtang@gzu.edu.cn

**Keywords:** defect detection, quality control, deep learning, object detection

## Abstract

The detection of product defects is essential in quality control in manufacturing. This study surveys stateoftheart deep-learning methods in defect detection. First, we classify the defects of products, such as electronic components, pipes, welded parts, and textile materials, into categories. Second, recent mainstream techniques and deep-learning methods for defects are reviewed with their characteristics, strengths, and shortcomings described. Third, we summarize and analyze the application of ultrasonic testing, filtering, deep learning, machine vision, and other technologies used for defect detection, by focusing on three aspects, namely method and experimental results. To further understand the difficulties in the field of defect detection, we investigate the functions and characteristics of existing equipment used for defect detection. The core ideas and codes of studies related to high precision, high positioning, rapid detection, small object, complex background, occluded object detection and object association, are summarized. Lastly, we outline the current achievements and limitations of the existing methods, along with the current research challenges, to assist the research community on defect detection in setting a further agenda for future studies.

## 1. Introduction

In the manufacture of mechanical products in complex industrial processes, defects such as internal holes [1], pits [2], abrasions [3], and scratches [4] arise, due to failure in design and machine production equipment as well as unfavorable working conditions. Products may also easily corrode [5] and be prone to fatigue [6] because of daily application. These defects increase the costs incurred by enterprises, shorten the service life of manufactured products, and result in an extensive waste of resources, thereby causing substantial harm to people and their safety [7]. Hence, detecting defects is a core competency that enterprises should possess in order to improve the quality of the manufactured products without affecting production. Automatic defect-detection technology has obvious advantages over manual detection. It not only adapts to an unsuitable environment but also works in the long run with high precision and efficiency. Research on defect-detection technology can reduce the production cost, improve production efficiency and product quality, as well as lay a solid foundation for the intelligent transformation of the manufacturing industry.

Therefore, many scholars have reviewed defect-detection-related technologies and applications to provide references for the application and research of defect-detection technology. For example, in view of the defect-detection technology applied by pharmaceutical products, Lalit Mohan Kandpal et al. [8]. summarized the application of hyperspectral [9], vibration spectrum [10], infrared [11], and other spectral technologies. For surface defect detection of manufactured products, Xianghua Xie [12] systematically recent advances in surface detection using computer vision and image processing techniques. By comparing the findings of past studies, they find that surface defect detection based on image processing requires high real-time performance in industrial applications. For fabric defect detection, scholars [13,14] reviewed the application and development of defect-detection methods commonly used in the production of textile fabrics from the perspective of defect-detection development of the textile industry production. Thermal imaging technologies are widely used in many industrial areas. I. Jorge Aldave [15] focused on the comparison of results obtained with commercially available non-experimental IR methods to provide references for the cameras in the field of non-destructive defect detection. Defect-detection technology is a hot topic in the industry and academia. However, scholars have yet to categorize product defect types (for example, steel [16] and textile [17]), the main detection techniques, summary of applications of defect-detection technology, existing equipment for defect detection, and other prospects. In addition, the mainline, review, and summary of the research status of relevant technologies locally and abroad have yet to be realized.

This paper first classifies the common defects of electronic components, pipes, welding parts, and textile materials, as shown in Figure 1. Then, it summarizes the mainstream deep-learning technology for defect detection and its application status and analyzes the application situation of the main defect-detection equipment, in order to provide reference for defect-detection technology in theory and practical application.

## 2. Survey of Defect-Detection Technologies

Product defect-detection technology is mainly to detect the surface and internal defects of products. The defect-detection technology refers to the detection technology of spot, pit, scratch, color differences and defect on the product surface. Internal defect-detection technology mainly includes internal flaw detection, hole detection and crack detection [30]. At present, several methods are used to detect product quality, including deep learning [31], magnetic powder [32], eddy current testing [33], ultrasonic testing [34], and machine vision [35] detection methods.

Wet magnetic particle detection mixes the magnetic powder in water, oil, or other liquid media. Magnetic powder marks the location of defects through liquid pressure and the attraction of the external magnetic field [32,36,37]. The moisture detection method has high sensitivity, and the liquid medium is recyclable [38,39]. Dry Magnetic powder testing [40] directly attaches magnetic powder onto the surface of the magnetized workpiece for defect detection. This method is used for the local inspection of defects in large casting, welding parts, and other segments that are unsuitable for wet detection. The continuous magnetic particle detection method detects defects in magnetic suspension or powder under the external magnetic field [41]. The method can be used to observe the defects in the external magnetic field. Several factors that influence the precision of Magnetic powder testing include roughness and the profile of the test piece, the geometrical characteristics of defects, the selected magnetization method, and the quality of operators [42]. Meanwhile, the factors that influence the sensitivity of Osmosis testing are imaging reagent, the performance of osmotic fluid, quality of operators, and the influence of defects. Factors that influence the accuracy of the detection of eddy current are the type and parameters of coil and material and the profile of the test piece [43,44].

The ultrasonic testing effect is affected by the angle between the defect surface and the ultrasonic propagation direction [34,45]. If the angle is vertical, then the signal returned is strong, and the defect is easily detected. If the angle is horizontal, then the signal returned is weak, which makes detecting a leak easy. Therefore, selecting the appropriate detection sensitivity and corresponding probe to reduce leakage detection is necessary [46]. The factors that influence ultrasonic testing include projection direction, probe effectiveness, sound contact quality, and instrument operating frequency [47,48].

Machine vision detection mainly consists of image acquisition and defect detection and classification. Because of its fast, accurate, non-destructive and low-cost characteristics, machine vision is widely used. Machine vision identifies objects mainly based on the color, texture and geometric features of objects. The quality of image acquisition determines the difficulty of image processing. In turn, the quality of the image processing algorithm directly affects the accuracy and error detection rate of defect detection and classification [49,50,51]. The deep-learning method is also a defect-detection method that is based on image processing, which is widely used to obtain useful features in massive data [52]. Table 1 presents a comparison of commonly used product defect-detection methods.

Clearly, the traditional defect-detection techniques and the popular deep-learning defect-detection techniques have their advantages. The traditional detection methods are highly focused. For instance, Osmosis testing technology [53] is only suitable for detecting defects in highly permeable and non-porous materials and have certain advantages over other general methods. However, most of the traditional detection methods still need to rely on manual assistance to complete, especially when a certain amount of instrument debugging is required before testing, and the equipment development cost is high, which is not highly adaptable and limited by the equipment life and manufacturing accuracy. Innovative defect-detection techniques, particularly machine vision and deep-learning methods [54,55,56], have become the most popular in recent years and are one of the key technologies for automating defect detection due to their versatility and lack of reliance on human assistance. Compared to traditional defect detection methods, the new technologies offer better inspection results and lower costs, but still rely on large amounts of learned data to drive model updates and improve inspection accuracy.

## 3. Survey of Deep-Learning Defect-Detection Technologies

Deep-learning technology has developed rapidly and made great success in object detection [61], intelligent robot [62], saliency detection [63], parking garage sound event detection [64], sound event detection for smart city safety [65,66], UAV blade fault diagnosis [67,68,69] and other fields [70,71,72]. Deep learning has a kind of deep neural network structure with multiple convolutions layer. By combining low-level features to form a more abstract high-level representation of attribute categories or features, the data can be better reached in abstract ways such as edge and shape to improve the effectiveness of the deep-learning algorithm [70], Therefore, many researchers try to use deep-learning technology to defect detection of product and improved the product quality [71,72,73,74]. Table 2 summarizes the advantages and disadvantages of deep-learning methods commonly used in product defect detection. It mainly includes convolutional neural network (CNN) [75], autoencoder neural network [74,76], deep residual neural network [77], full convolution neural network [78], and recurrent neural network [79,80].

(1) Using the CNN to defect detection of product [75]. CNN is a feedforward neural network. CNN consist of one or more convolutional layers and fully connected layers, as well as associated weights and pooling layers [81]. Literature [82] is a very popular LeNet convolution neural network structure. LeNet network structure can be used to detect defects in two situations: one is to design a complex multi-layer CNN structure, use different network structure to extra image content features, and complete end-to-end training to detect defects in images [56,83]; the other is to combine CNN with CRF model, train CNN with CRF energy function as constraint or optimize network prediction results with CRF. And to achieve the detection of product defects [71].

(2) The product defect-detection technology based on the neural network [74,76]. Autoencoder network mainly includes two stages: coding and decoding. In the coding stage, the input signal is converted into a coding signal for feature extraction; in the decoding stage, the feature information is converted into a reconstruction signal, and then the reconstruction error is minimized by adjusting the weight and bias to realize the defect detection [84]. The difference between autoencoder networks and other machine learning algorithms is that the learning goal of the autoencoder network is not for classification, but for feature learning [85,86]. It also has a strong ability of autonomous learning and highly nonlinear mapping. It can learn nonlinear metric functions to solve the problem of segmentation of complex background and foreground regions [87].

(3) The product defect-detection technology of deep residual neural network [77]. The deep residual network adds a residual module on the basis of the convolutional neural network. The residual network is characterized by easy optimization and can improve the accuracy by increasing the network depth [88,89]. CNN, Generative Adversarial Networks [90], etc. with the depth of the network increases, the extraction feature increases, but it is easy to cause the activation function not to converge. The purpose of the deep residual network is to optimize the increasing number of network layers with residual while increasing the network structure, so that the output and input element dimensions of the convolution layer in the residual unit are the same, and then through the activation function to reduce the loss.

(4) Full convolution neural network [78]. The fully connected layer is a connection between any two nodes between two adjacent layers. A fully connected neural network uses a fully connected operation, so there will be more weight values, which also means that the network will take up more memory and calculations [91]. During the calculation of the fully connected neural network, the feature map generated by the convolution layer is mapped into a fixed-length feature vector. The full convolution neural network can accept the input image of any size, and use the deconvolution layer to sample the feature map of the last convolution layer, it can recover to the same size of the input image.so that a prediction can be generated for each pixel, while retaining the spatial information in the original input image, and finally classify the feature map of the upper sampling pixel by pixel.

(5) Recurrent neural network recursively from the evolution direction of sequence data and all cyclic units are connected in a chain manner, and the input is sequence data [79,80]. The CNN model mainly extracts the feature information of input layer test samples through convolution and pooling operations. The recurrent neural network uses the recurrent convolution operation to replace the convolution operation on CNN. The difference is that the recurrent neural network does not perform the pooling layer operation to extract the features after the recurrent operation to extract the input layer features, but uses the recurrent convolution operation to process the features of the samples.

## 4. Survey of Object Detection Technologies Based on Deep Learning

Object detection methods based on neural networks can be divided into a one-stage method based on regression [92,93,94,95,96] and a two-stage method based on candidate box generation and classification [97,98,99,100,101,102,103]. The one-stage method does not need to generate candidate boxes in advance, but it only needs to complete the three tasks of feature extraction, classification, and location regression. By contrast, the two-stage method mainly has four tasks, namely feature extraction, generating candidate boxes, classification, and location regression. Table 3 shows a comparative analysis of the two types of object detection methods.

Representative one-stage methods include: Joseph Redmon et al. [92,93]. proposed the You Only Look Once(YOLO) method, which inherits OverFeat, and its detection speed reaches 45 pieces per second. The speed advantage makes it an end-to-end leader. Redmon et al. [94]. modified the network structure of YOLO and proposed the YOLOv2 and YOLO9000 methods, with a 4.00% increase in mAP. YOLOv3 [95] follows the Darknet53 network of YOLOv2 and combines it with the FPN [96] network structure. Thereafter, the prediction results are obtained from the convolutional network. Corresponding improvements enable the accuracy to reach 22.2 milliseconds per piece, and the best effect of mAP@0.5 on the COCO test set reaches 33.00%. However, the overall model becomes considerably complicated, and speed and accuracy serve as checks and balances for each other. To solve the problem of poor positioning accuracy of the YOLO algorithm, Wei Liu et al. [97]. proposed the Single Shot MultiBox Detector(SSD) method that combines YOLO regression ideas with the Faster R-CNN [98] anchor box mechanism, using the Visual Geometry Group Network(VGG) as the feature extraction network. The VGG is a convolutional neural network model proposed by K. Simonyan [99]. To address the issue that the SSD algorithm cannot easily detect small objects, Cheng-Yang Fu et al. [100]. proposed a DSSD method, which replaces the VGG16 of SSD with the ResNet101 network, thereby enhancing the network’s ability to extract features.

The representative two-stage methods are as follows. Girshick et al. [101]. proposed the Rich Feature Convolutional Neural Network (R-CNN) to enrich the features in the training process. That is, the mAP of the PASCAL VOC2007 test set was refreshed to 58.50%. He et al. [102]. proposed the Spatial Pyramid Pooling Network(SPP-Net) algorithm based on R-CNN to solve the problems of repetitive operations and shape distortion of convolutional neural networks. SPP-Net abandons the R-CNN clipping candidate box and image sub-block scaling operations before inputting the neural network, and adds an SPP structure between the convolutional and fully connected layers to increase the generation rate of candidate box and save computational overhead. Given the time cost caused by repeated computation of multiple stages and features during training and the space cost caused by storage of intermediate feature data, Girshick et al. [103]. proposed Fast R-CNN, which combines deep network with SVM classification. Accordingly, classification and regression are performed simultaneously by the full connectional layer network, thereby forming a multi-task model. This module has numerous calculations, given the problem that SPP-Net and Fast R-CNN have separate candidate area modules. Ren et al. [27] proposed the Faster R-CNN algorithm based on Fast R-CNN. Faster R-CNN adds an RPN network to the backbone network structure and sets multi-scale anchor points on the basis of established rules. To solve the problem that the Faster R-CNN rounds the feature map size when performing ROI pooling and downsampling, Mask R-CNN [104] abandons the rounding operation of the picture size and proposes to replace the ROI Pooling layer with ROI Align and use double Linear interpolation fills pixels at non-integer positions. Accordingly, no position error occurs when the downstream feature map is mapped upstream.

## 5. Summary Analyses of the Application Status of Defect-Detection Technology

### 5.1. The Traditional Method for Defect-Detection Technology

Non-destructive defect detection of products is widely used in manufacturing, in which analyzing the pros and cons of different algorithms can help to understand and improve the algorithms. Here we focus on the application status using the combination of classical defect detection and other algorithms. Figure 2. shows the different defect-detection methods and their corresponding performance results or summaries for non-destructive defect detection.

The ultrasonic defect-detection methods are widely used to detect the defects in the internal structure of the sample. Therefore, the results are mainly reflected in the performance of the ultrasonic signal [105]. The findings, as shown in [106], indicates that the ultrasonic defect-detection methods have the advantages of fast detection speed and simple operability. They also have special advantages in detecting defects in the internal material and structure as well as the size of the product. However, this method is unsuitable for workpieces with complicated structures with low detection efficiency [106]. Ultrasonic techniques are especially ineffective for detecting defects on the upper surface of the sample since a nonlinear relationship exists between the defect position and the signal receiving the time, which leads to the defect to be closely positioned to the direct pass wave end [107]. The denser the distribution of the real position of the product, the higher the certainty of the “trailing” phenomenon of the direct pass wave signal on the map.

The machine vision-based defect-detection methods are suitable for the detection of surface defects in products, which has achieved up to 88.60% accuracy in binary defect-detection problems [108]. The defect-detection accuracy over scratches, holes, scales, pitting, edge cracks, crusting, and inclusions can reach 95.30% [109]. Our survey shows that the machine vision-based image recognition is widely used in manufacturing surface defect detection due to the feature extraction ability from the images made possible by recent deep-learning techniques.

The defect-detection methods are based on filtering has a strong ability to describe the disturbance signal and detection of the tool defect inside the machine.

To the abovementioned main categories of defect-detection methods for mechanical products, several other technologies are also available such as the X-ray image defect-detection technology [110], Pulse magnetoresistance method [111], Acoustic emission technology [106]. These methods [106,110,111] have shown positive detection results and can provide theoretical and practical guidance for real-world applications. In addition, studies show that the majority of the early works studied a single defect-detection problem such as the defects in the material [112,113], shape [114,115], size [116], color [117], and surface of a product [118,119]. At the same time, studies on defects with varying size, crack depth, and other information are scarce, which is also a major limitation of existing defect-detection research [120,121,122]. The following is an experimental summary of some research methods [108,109,110,123,124,125,126,127,128,129,130,131,132]. The reference [123,124,125,126] are the experimental results of ultrasonic detection. The reference [109,127,128,129,130,131] are the experimental results of Filter detection. The reference [108,110,132] are the experimental results of other quality detection technologies.

The frequency accuracy of the weld defect is over 60.00% when the frequency band is 100–200 kHz.Positive detection results can be obtained at frequencies of 100 to 200 kHz and 300 to 400 kHz [106].The total detection rate of the two types of samples in “defects” and “no defects” is 88.60% [108].Analysis of the effects of different surface defects and locations on the test results.The average recognition rate under eight types of defects is 95.30% [109].Poor detection effect with defect depth less than 2mm [110].The peak times of surface and subsurface defect depth of 3 mm are 16.59 and 37.01 ms, respectively [111].Realizes the detection and recognition of defects in different texture samples [123].Reduces unnecessary interference and extract weak signals from strong background noise [124].Effective detection of holes, axial cracks, and circumferential cracks [125].Detect cracks less than 3 mm [126].Can detect defects exceeding 1 mm^2^ [127].Resolves the quality defect detection with image noise and complex background [128].Improved Doppler distortion and multi-bearing source aliasing in bearing signals [129].Positive recognition effect on the position, shape, and size of the defect [130].It can define six features based on the characteristics of seam cracks and employed SVM for classification. The true positive rate was 94.46%, and the false-negative rate was only 0.29% [131].The 8-bit grayscale image recognition rate of an image size of 2500 × 2000 pixels is 94.00% [132].

### 5.2. Machine Learning for Defect-Detection Technology

Another major trend in the literature survey of defect-detection technology is the emerging dominance of the machine learning methods, which are now widely used in all fields of product defect detection. The defect-detection technology can be divided into two main categories: surface defect detection [133] and internal fault diagnosis [134]. Surface defect detection is similar to ’visual’ detection, that is, learning from the target features in an image with the help of deep-learning image processing technology to classify and locate product defects in the image, whereas internal fault diagnosis is similar to ’Auditory’ detection, that is, the diagnosis of faults in rotating parts such as bearings by means of modal analysis using digital signals in the time or frequency domain. We found that the defect-detection complex functions and enhanced feature extraction [135]. Table 4 Survey of Deep-learning methods in defect methods based on deep learning have achieved the best experimental result thus far. The highest precision of these algorithms can reach 99.00% with a recognition time of 0.12 ms for a single image [136]. The lowest precision level is 86.20% [137].

## 6. Survey of Defect-Detection Equipment

At present, defect-detection equipment is mainly used in the production and processing stage of products. In the 1960s, due to the demand for the medical examination market, machine vision-based defect-detection robots are developed to detect insoluble foreign bodies in medical injection. Many visual inspection equipment providers were born such as BOSCH(Germany) [151], COMPUR [152], BS [153], CMP (Italy) [154], and Valley Industries (Japan) [155]. Recently, demand for intelligent manufacturing has been increasing as shown in China’s intelligent manufacturing 2025, German industry 4.0 and Britain’s “Modern Industrial Strategy”. Driven by these practical applications, demand for defect-detection equipment has been increasing steadily.

Figure 3 presents the defect-detection equipment widely used in the industry. In Table 5, we summarize the advantages and disadvantages of these defect-detection equipment aiming to identify the recurrent and difficult challenges in defect detection. Figure 3a features a defect-detection system for LYNX mechanical parts, which detects missing mechanical parts and external surface damages and assembly of parts. Figure 3b shows the use of machine vision to visually detect component defects. However, it is found that current detection accuracy and detection performance remain incapable of fully meeting the requirements of high-speed production in smart factories. Furthermore, stability and real-time performance should be further improved.

## 7. Challenge

### 7.1. 3D Object Detection

In modern computer vision systems, people are satisfied with 2D object detection. With the advent of 2.5D depth sensor, building 3D models has become increasingly important, and 3D geometric shapes have been considered an important clue in object recognition. One of the most important challenges of 3D is that given a depth map of an object from an angle, the 3D structure of an object can be inferred. We are convinced that one of the important trends in visual recognition is to identify defects on the surface of an object and infer its corresponding 3D model to determine the shape defects in the 3D layer. This situation is akin to that when we observe one side of a table, i.e., we naturally visualize a 3D model of the entire table. To determine the table based on the surface defects and in the 3D shape of the defects, various methods can be used to reconstruct and synthesize shapes, although such techniques are based on assembly. They are the same class or type of 3D shape completion. The completion of 3D models of different types of objects in complex environments should be studied further. At present, similar studies have been conducted. For example, the linear relationship between the 3D defect size of transparent parts and image gray level is found under certain circumstances [164]. On this basis, a set of vision-based transparent micro-defect measurement systems is developed. Iglesias [165] developed an automated inspection system to examine SLATE based on the use of 3D color cameras to capture data and using computer vision algorithms developed specifically for the purpose of studying SLATE characteristics. In a previous study [166], a potato virtual reality model reconstruction algorithm based on 3D shape and color images is proposed for sample quality tracking and review. The model redisplays potato color and 3D shape data in multiple views and supports 360-degree rotation in horizontal and vertical directions to simulate a handheld exam experience. This depth image processing is a potentially effective method for future non-destructive post-harvest grading, particularly for products in which size, shape, and surface conditions are important factors. In a previous study [167], a detection method combining grayscale image and 3D depth information is proposed.

### 7.2. High Precision, High Positioning, Fast Detection, Small Object

Through investigation, it is found that high precision [96,98], high positioning [94,96,98], fast detection [94], small object, complex background [94,96,98,168], occluded object detection [94,168], and object detection based on the association between objects [96,98] are the main challenges of the current deep-learning algorithm in the application of quality detection. In order to further assist researchers and enterprise engineers to apply the deep-learning method to product defect detection, Table 6 are from high precision [96,98], high positioning [94,96,98], fast detection [94], small object [96,100], complex background [94,96,98,168], occluded object detection [168,169], object association [170], and other aspects are summarized in the relevant papers published in ICCV and CVPR and other well-known international conferences in recent years, and the core idea and source code of these better papers are summarized to assist R & D personnel in knowledge reuse and innovative design.

## 8. Development Trend

Our survey found that most of the existing defect-detection studies focused on defect detection of specific products. But the identification accuracy of current online defect-detection techniques remains to be improved. The following aspects of defect detection need special attention:Combined with the actual requirements of the factory, online defect detection of manufacturing products on the conveyor belt should be realized.As intelligent manufacturing enterprises attach importance to defect-detection technology, embedded sensor equipment to conduct online real-time detection of defects in manufactured products can be designed and used. Then, various non-destructive defect-detection methods should be integrated to realize multi-modal defect detection of manufacturing products, which can have broad application prospects in the field of defect detection.The main objects of 2D image surface defect-detection technology are surface scratches and abrasions. Obtaining in-depth information about the defects is limited. However, in the actual production process, the defect information of the product is not only displayed on the surface of the manufactured product but also requires the use of 3D defect-detection methods to detect the 3D surface characteristics of the test sample.With the rapid development of artificial intelligence and big data technology, useful information that can be extracted is abundant. Applying the rich information accurately to the manufacture of product defect feedback technology, defect control, and fault diagnosis warrants further research.Aiming at the multi-fault diagnosis of intelligent equipment with defect-detection technology in complex industrial processes, one of the important research directions to undertake should be effective fault prediction and diagnosis for intelligent equipment when multiple faults simultaneously occur.High-precision identification technology. In the process of image acquisition, the apparent characteristics of the object can considerably change with different lighting conditions and shooting angles and distance. Many noise interference and partial occlusion of the detected sample can also have a great impact on the detection results due to the different backgrounds of the detection object. The abovementioned factors are commonly used in industrial applications, which can lead to substantial difficulties in detecting defects in manufactured products. Therefore, such a problem should be further solved to improve the feature extraction capability of the current online non-destructive defect-detection technology and improve the accuracy of non-destructive defect detection.How to optimize the quality of image acquisition, improve the accuracy of the candidate box, extract features more comprehensively and accurately for learning, and extract features of small-size targets are the future research directions;Presently, a large number of neural networks (including neural networks improved for a certain problem) have their own advantages and disadvantages. These networks are implemented based on a large amount of data. How to use fewer picture samples to train the recognition model with excellent performance is a big difficulty;With more and more product derivatives, it remains to be studied how to transfer a trained model to another similar product and ensure its accuracy and detection efficiency;After the defect is detected and the type of defect is clear, it is very important to deal with the information of the object, and it is also very necessary to separate the defective product from the non-defective product. The defect-detection system can be combined with the early warning system to give timely warning after detecting the defective products, and the staff can timely eliminate the defective products. Or with the sorting system, the manipulator to eliminate the defective products, in addition, can also establish traceability system to check the production process will make the product defects steps, and timely optimization of the production process, so as to reduce the production cost;Future studies can also design defect information feedback technology that is based on defect-detection technology. Many feedback methods and objects remain undiscussed and are difficult points for future research. Once product information is processed and analyzed, and after determining the cause of the product defect or fault information, the defect or fault information can be fed back to the mother-machine system to realize online production and self-correction of the product. Doing so can help improve product quality and reduce manpower, material resources, and production costs. Finally, we hope to compare the performance of the mainstream deep-learning detection model, which can provide a reference for researchers in deep-learning surface defect detection. See Table 7 for details.

## 9. Summary

Industrial product quality is an important part of product production, and the research on defect-detection technology has great practical significance to ensure product quality. This article provides a comprehensive overview of the research status of product defect-detection technology in complex industrial processes. We have compared and analyzed traditional defect-detection methods and deep-learning defect-detection techniques, and comprehensively summarized the experimental results of defect-detection techniques. Meanwhile, combined with the actual application requirements and the development of artificial intelligence technology, the defect-detection equipment was investigated and analyzed. Through investigation, we found that 3D object detection, high precision, high positioning, rapid detection, small targets, complex backgrounds, detection of occluded objects, and object associations are the hotspots of academic and industrial research. We also pointed out that embedded sensor equipment, online product defect detection, 3D defect detection, etc. are the development trends in the field of industrial product defect detection. We believe that the investigation will help industrial enterprises and researchers understand the research progress of product defect-detection technology in the field of deep learning and traditional defect detection.

## Figures and Tables

**Figure 1 materials-13-05755-f001:**
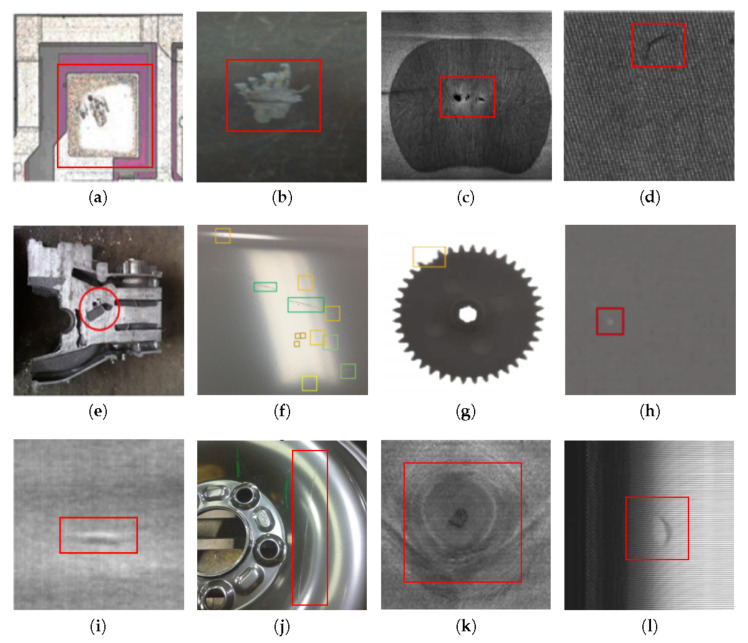
Defects in different areas: (**a**) metallization peeloff of electronic components [18]. (**b**) pipeline corrosion [19]. (**c**) defective with gas pore [20]. (**d**) defect bigknot of textile materials [21]. (**e**) shrinkage and porosity defect of Casting [22]. (**f**) defects in green, yellow, orange bounding box are scratch, cratering, hump, respectively in carbody [23]. (**g**) Lack defect of gear [24]. (**h**) light leakage defect on mobile screen [25]. (**i**) Convexity defect in aluminum foil [26]. (**j**) Scratch defect of the wheel hub [27]. (**k**) Branch defect of wood veneer [28]. (**l**) Bubble defect of tire sidewall [29].

**Figure 2 materials-13-05755-f002:**
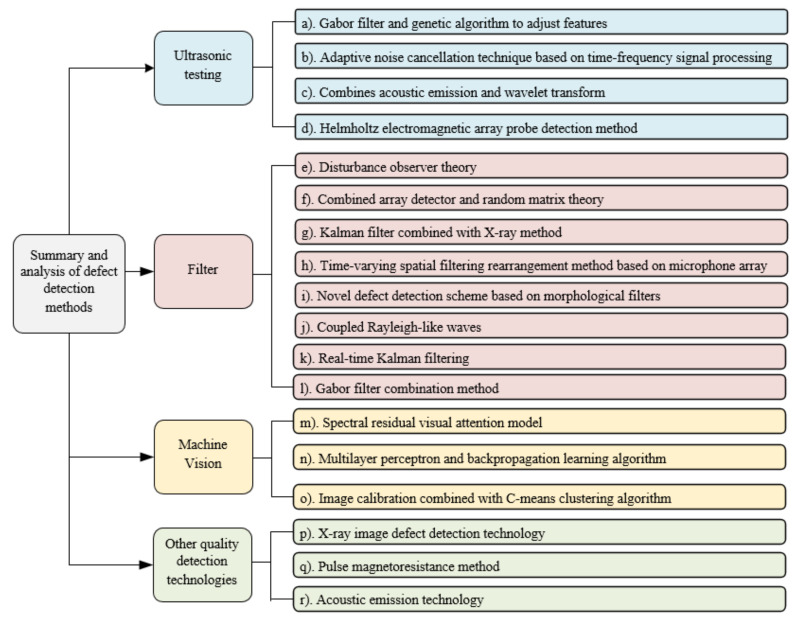
Summary and analysis of defect-detection methods.

**Figure 3 materials-13-05755-f003:**
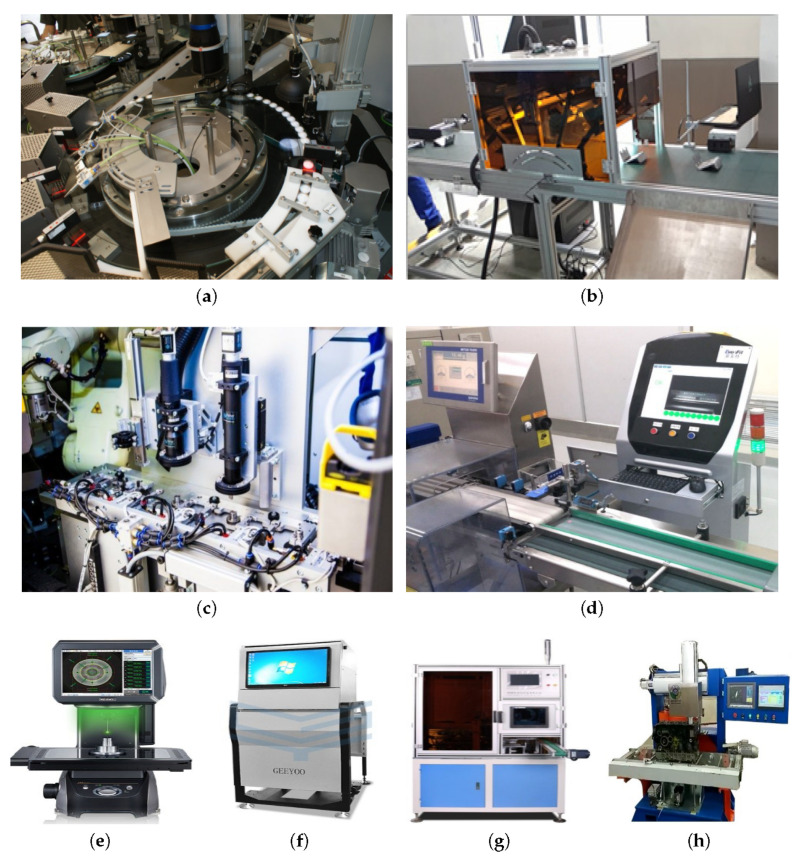
Types of defect-detection equipment: (**a**) Rhein–Nadel automation (RNA) glass defect-detection system [156] (**b**) Visual detection system for defects of EvenFit components [157] (**c**) Rhein–Nadel automated detection system for defects in mechanical parts [158] (**d**) The visual detection system of the EvenFit capsule [159] (**e**) KEYENCE product size measuring instrument [160] (**f**) AVI soldering appearance inspection machine [161] (**g**) Hardware and workpiece visual measurement equipment [162] (**h**) ET-F1 engine cylinder bore eddy current detector [163].

**Table 1 materials-13-05755-t001:** Comparison of common defect-detection methods.

Methods	Strengths	Weaknesses	Applicable
Ultrasonic testing [45]	Easy to use, strong penetration, high sensitivity, portable equipment, and automatic detection.	Unsuitable for complex workpieces.	Any material
Machine vision detection [51]	A wide range of applications, high precision, remains unaffected by the profile of the detection piece, and automatic detection.	Detects surface defects only.	Any material
Magnetic powder testing [57]	The position, shape, and size of the defect can be visualized, which is suitable for any size of workpiece detection. It has the characteristics of high precision and low cost.	Application is limited to ferromagnetic materials. Detection results are affected by the geometric shape of the test pieces. Realizing automatic detection is difficult.	Ferromagnetic materials (e.g., cast steel, pipe, calendar, bar, etc.)
Osmosis testing [58]	Free from the influence of material type and shape profile and high sensitivity to pinhole defects.	Detecting porous materials is difficult, and the detection speed is slow. Detection results are greatly affected by the inspectors, and automatic detection is difficult to carry out.	Nonporous materials are tested (e.g., metal casting, ceramic, plastic, glass, etc.)
Eddy current testing [59]	Noncontact detection, fast detection speed, high sensitivity, and suitable for high-temperature environments, automatic detection.	The shape and size of the defects cannot be visualized. The applicable materials are limited. Difficulty in detecting deep defects with low detection accuracy.	Conductive or non-metallic material (e.g., workpieces, pipes, wires, and graphite)
X-ray testing [60]	Non-destructive detection, strong penetration, free from the influence of material appearance and structure, and easy operation.	Radiation effects for the staff involved in the detection.	Any material

**Table 2 materials-13-05755-t002:** Deep-Learning Defect-Detection Methods.

Methods	Strengths	Weaknesses	Applicable
CNN	It has a strong learning ability for high-dimensional input data and can learn abstract, essential and high-order features from a small amount of preprocessed and even the most original data.	The good expression ability and the calculation complex will increase with the increase of network depth.	Unlimited material
Autoencoder neural network	It has a good object information representation ability, can extract the foreground region in the complex background, and has good robustness to the environment noise.	The input and output data dimensions of the autoencoder machine must be consistent.	Unlimited material
Depth residual neural network	The residual network has lower convergence loss and does not overfit, so it has better classification performance.	The network must cooperate with deeper depth to give full play to its structural advantages.	Unlimited material
Full convolution neural network	It can extract the feature of any size image, and obtain the high-level semantic prior knowledge matrix, which has a good effect on semantic level object detection.	The feature matrix transformation combined with the underlying features is needed, and the convergence speed of the model is slow.	Unlimited material
Recurrent neural network	When there are fewer sample data, we can learn the essential features of the data and reduce the loss of data information in the process of pooling.	With the increase of the number of iterations in the network training process, the recurrent neural network model may appear overfitting phenomenon.	Unlimited material

**Table 3 materials-13-05755-t003:** Comparative analysis of two kinds of object detection methods.

Methods	Onestage Object Detection	Twostage Object Detection
Principle	The input original image is processed directly to obtain the position coordinate value and category probability. The position is corrected thereafter.	Candidate regions are extracted from the input image through selective search and region generation network. Thereafter, convolution, pooling, and other processing is conducted to obtain feature maps.
Advantage	In the case of the low input separation rate, the speed and accuracy can be balanced simultaneously, and the detection speed is fast, which can reach above 45 FPS.	The deep semantic features of the object can be obtained. The detection accuracy of the object is high, whether it is a small object or a scene with considerable density.
Insufficient	Low accuracy for small objects and prone to miss detection, low positioning accuracy.	The algorithm has a large volume, large amount of stored data, complicated calculation process, and slow detection speed.
Realtime	Realtime.	Cannot reach real time

**Table 4 materials-13-05755-t004:** Machine Learning for Defect-Detection Technology.

**No.**	**Methods**	**Performance**
1	Deep ensemble learning [23]	Recall of 93.00% and detection precision of 88.00%.
2	Deep CNN [42]	The proposed fast architecture mAP is 96.72%, whereas FPS is 83.00 Training time consumption is 133 min.
3	CNN [136]	The overall recognition rate of the six kinds of defect dataset reaches 99.00%. The recognition time of a single image is 1.2 ms.
4	Deep CNN [137]	The CNNs were trained using 12,000 images that were collected from over 200 pipelines. The average testing accuracy, precision, and recall rates were 86.20%, 87.70%, and 90.60%, respectively.
5	CNN and Naïve Bayes data fusion [138]	The Naïve Bayes decision making discards false positives effectively. The proposed framework achieves a 98.3% hit rate against 0.1 false positives per frame.
6	Machine learning [139]	The ICA, Gabor filter, and RF require approximately 0.097, 0.265 and 0.014 s, respectively, to detect defect for a 640 × 480-pixel image with sliding window search. However, CNN takes 0.217s and is slower than ICA and RF.
7	CNN and self-similarity [140]	Benchmarked on a publicly available dataset of SEM images, outperformed the state of the art by approximately 5% by reaching an area under the curve of approximately 97.00%.
8	3D active stereo [141] omnidirectional vision sensor	The highest accuracy to detect defects is 97.00%. The recognition time of a single image is 0.19 s.
9	Deep neural networks [142]	This paper obtained a mean IOU of 68.68% over 55.94%. The performance of all three metrics on the validation data reflect the superiority of adversarial training.
10	Deep CNN [143]	Maximum accuracy of the 32 × 32 pixel-sized image is 94.68% in industrial detection.
11	Support vector machine and CNNs [144]	In-wheel defect detection, accuracy is larger than 87.00%. The precision value is larger than 87.00%. The recall rate is larger than 89.00%.
12	Deep convolutional autoencoder [145]	Recall 95.70% Precision 91.80% the inspection time for an image of 512 × 512 pixels is only 20 ms.
13	CNN [146]	Average accuracy of 93.02 % only 8.07 ms for predicting one image on an ordinary computer.
14	CNN [147]	The mean accuracy of 99.38% with the std value of 0.018.
15	Fully Convolutional Neural Network [148]	Accuracy 99.14% a batch of 50 images required only 0.368 s.
16	Machine learning [149]	Accuracy as high as 99.4%.
17	Few-shot Learning [150]	Accuracy rate can reach 97.25%.

**Table 5 materials-13-05755-t005:** Existing defect-detection equipment.

**Name**		**Performance**
Packaging defect-detection equipment	Function	color detection, window or insert detection, carton ejection, tilt, double-feed monitoring and glue line detection of mechanical product packaging.
	Trait	remote control, off-site monitoring, and tracking. ineffective for metal or special transparent packing.
LYNX Industrial vision system	Function	detect and analyze missing and damaged parts and assembly errors.
	Trait	System hardware can be controlled by the central terminal, which can adapt to various working environments and cover the detection amount of size and size. It is easy to operate, fully closed, a multi-detection system with data.
IRNDT infrared thermal imaging testing	Function	The defect image is displayed, and the feature is evaluated by heating the tested parts and analyzing the defect position with abnormal internal temperature.
	Trait	The equipment has the characteristics of the non-contact, large area, fast speed and visual display it has a poor effect on metal parts with a large amount of heat deformation.
Smart U32 Ultrasonic scanning detector	Function	This is used to perform ultrasonic phased array probe, sound beam control, and dynamic focusing technology to realize scanning and imaging detection of composite and metallic materials.
	Trait	Coupling stability, automatic measurement, and accurate verification. However, it is not good for the detection of large-size parts and non-metal parts.
Parts appearance optical detection equipment	Function	This is used to detect parts with diameters ranging from 65 mm to 110 mm, such as sprockets, stators, and rotors. It can also detect breakage, bumps, and cracks.
	Trait	TPros: non-destructive testing, multi-angle identification, fast detection speed, high precision, stable performance, and accurate data statistics function.
Turbine detection system	Function	The defect of the shallow surface of metal parts is detected through the analysis and treatment of the eddy current. It is suitable for defect detection of conductive materials.
	Trait	Eddy current testing is only applicable to conductive materials. It can only detect defects on the surface or near the surface layer. It is not conducive for use in components with complicated shapes.
Sealing detection equipment	Function	It collects and analyzes the images of the seal ring directly above, sides, and bottom and extracts the surface scratches and bubble defects of the seal ring.
	Trait	The equipment can set the number of test stations, adjust the test sequence and methods at will, and support the detection of all product models. The detection speed is slow, which reduces the production tempo along the pipeline.
3D Visual measuring equipment	Function	It has the functions of edge extraction, contour degree and other 2D and 3D form tolerance calculation, 3D digital model comparison, and heat map display.
	Trait	The device can only detect non-transparent products, and the measuring effect is insufficient when the running speed is over 400 mm/s.
Inkjet detector	Function	It can detect defects in mechanical parts without and incomplete codes, indented characters, and offset position of the code.
	Trait	The device consists of a code detection unit, man-machine interface, and stripper. However, it is ineffective in detecting parts with greasy surfaces.

**Table 6 materials-13-05755-t006:** Summary of object detection methods with high precision, high positioning, fast detection, small object, complex background, occluded object detection.

Ref.	High Precision	Position Ability	Fast	Small Object	Train Strategy	Irregular Object	Imbalance Data	Complex Background	Occluded Objects	Objects Relationship	Published
[94]		✓	✓		✓			✓	✓		CVPR2017
[96]	✓			✓	✓					✓	CVPR2017
[98]	✓	✓			✓		✓	✓			CVPR2017
[100]			✓	✓	✓			✓	✓		arXiv
[104]	✓				✓			✓	✓		ICCV2017
[168]								✓	✓		ICCV2017
[169]					✓				✓		ICCV2017
[170]				✓						✓	CVPR2018
[171]	✓					✓		✓			CVPR2017
[172]	✓				✓						ICCV2017
[173]						✓		✓			ICCV2017
[174]					✓		✓				ICCV2017

**Table 7 materials-13-05755-t007:** Performance comparison of deep-learning object detection model.

Model	Network Structure	Real-Time Performance Analysis	mAP		Published
			VOC2007	COCO	
Faster R-CNN [27]	ResNet101	Poor realtime performance	73.20%	37.40%	NIPS’15
YOLO [92]	VGG16	Good real-time performance	66.40%	23.70%	CVPR’16
OverFeat [93]	−−	Poor real-time performance	−−	−−	ICLR’14
YOLO V2 [94]	Darknet19	Good real-time performance	78.60%	21.60%	CVPR’17
YOLOv3 [95]	Darknet53	Good real-time performance	−−	57.90%	Arxiv’18
FPN [96]	ResNet	Good real-time performance	−−	36.20%	CVPR’17
SSD [97]	VGG16	Good real-time performance	76.80%	31.20%	ECCV’16
DSSD [100]	ResNet101	Good real-time performance	81.50%	33.20%	Arxiv’17
R-CNN [101]	AlexNet	Poor realtime performance	58.50%	−−	CVPR’14
SPP-Net [102]	ZF-Net	Poor realtime performance	59.20%	−−	ECCV’14
Fast R-CNN [103]	VGG16	Poor realtime performance	70.00%	19.70%	ICCV’15
Mask R-CNN [104]	ResNet101	Poor real-time performance	−−	39.80%	ICCV’17
R-FCN [171]	ResNet101	Poor realtime performance	79.5%	29.90%	NIPS’16
MegDet [175]	ReseNet	Good real-time performance	−−	52.50%	CVPR’18

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
