# Peer review of "Using Deep Learning to Detect Defects in Manufacturing: A Comprehensive Survey and Current Challenges"

_materials, 2020, doi:10.3390/ma13245755_

Round 1
Reviewer 1 Report
In this paper the authors reviewed deep learning methods in defect detection. This article needs improvement in all sections. This is an article that was supposed to analyze the techniques for identifying defects and to compare them. First of all, therefore, all the techniques for identifying defects should be properly introduced. These techniques should be explained in detail as this is a survey. Unfortunately, from this point of view the article is weak. Therefore, it is necessary that the authors dedicate themselves in greater detail to the explanation of both traditional and innovative defect identification technologies. It is not enough to enter the method definition and then add a reference. As for methods based on deep learning, they are only often mentioned with acronyms without specifying what they are. This way the reader will not be able to understand how these methods are used to identify defects. Throughout the paper massive use is made of tables and figures without commenting them adequately, leaving it to the reader to analyze them. Also I think it is necessary to check the copyright of many of the images included in the paper.
17-19) It would be advisable to specify which production field you are referring to.
23) mechanical products. Ok now you do, move this clarification to the beginning.
25-26) maybe you meant automated Defect detection technology?
33) et al[8]. move the period
44) don’t use abbreviations
48-49) So not just mechanical parts? It is necessary to specify precisely what kind of products this study refers to. This clarification must be made at the beginning.
51)” reference and reference” ?
52)move table 1 after the Figure 1, since it calls you back to line 83
52) you should reformat Table 1, in the Weaknesses column you should sort the points as a numbered list
52) In Figure 1 there are images extracted from several articles, do you have permission to publish them?
54) Only now define the Product defect detection technology, it would be appropriate to do it in the previous section. This way the reader will know immediately what you are referring to.
58-68) Add more references to allow the reader to deepen the topics.
85-86) Add more references to problems addressed by deep learning-based model to allow the reader to deepen the topics. For example, "Parking Garage Sound Event Detection", and "Sound Event Detection for Smart city safety"
90-91) Add more references to defect detection problems. For example, “UAV blade fault diagnosis”.
92) Fig. 2 is LeNet network structure, have you the permission for publish? Unfortunately, citing the source is not enough.
93) Adequately introduce this list of technology you are proposing
93)Introduce CNN, not all readers are necessarily machine learning experts
100-107)Add reference to Autoencoder network, so the reader can deep the topic
108) Introduce deep residual neural network
115) the capital letter is missing
115-121) Re-word the whole section, it seems that something is not clear. First explain the Full convolution neural networks.
128) Fig. 3 and 4. Have you the permission for publish?
130-131) This classification is not clear to me, can you add some references to support your statement?
137) Introduce adequately the YOLO method, also add the complete definition maybe it makes you understand something more - YOLO (You Only Look Once)
143)” COCO test-dev” Explain
146)SSD, add the complete definition and introduce the topic
147) VGG16, introduce the topic
153) SPP-Net and R-CNN, introduce the topics
179-181) Were I can find this conclusion? Explain better this statement, and not to say that these conclusions are obtained from Fig. 6
189) Why second? Fig. 6 shows the filter as second.
191) The reference 84 doesn't seem to me a traditional method
199-201) Ad references for theoretical and practical guidance
201-203) There seems to be something wrong here. Table 3 refers to Comparative analysis of two kinds of object detection methods
205-220) I don't understand this listing, what it refers to. Before proposing a list of arguments you should introduce it. What are you proposing? Reading the sentences it seems that you have extracted two words to quote a work, in this way the reader will no longer be able to follow the flow of information.
227) Table 5 refers at other topic
234) In Table 4, distinguish between machine learning and deep learning. Deep learning is a subfield of machine learning.
234-235) Inserting abbreviations is not good. I think it is a company then you should at least insert a reference
239) Use the same notation for the entire paper to call up the figures
241) Use the same notation for the entire paper to call up the figures
243) Use the same notation for the entire paper to call up the figures
246) In Fig.7 caption the first letter is missing. Where did you get these images from? Cite the source.
274-282) Add references
Author Response
Response Letter
Dear Editors and Reviewers,
Thank you for your letter and for the reviewers’ comments concerning our manuscript entitled “Using Deep learning to Detecting defects in manufacturing: A Comprehensive Survey and Current Challenges” (ID: materials-1006601). We are very excited to have this opportunity to revise our manuscript. Those constructive comments are valuable and very helpful for improving our paper and for guiding our future research. We want to extend our appreciation for taking the time and effort necessary to provide such insightful guidance. Mainly made the following changes。
- The paper was revised according to the Reviewer.Specific modification information cover letter.
- Polish the language;
- Deleted pictures that may cause copyright disputes;
- Added references 42-48,58-61,65-70,77,82-86,101,107,109,115-125,140,145-156;
- All revised parts in the paper have been marked in red.
- Added description of recurrent neural network.
We now detail our responses to each of the reviewers’ concerns and comments.
We tried our best to improve the manuscript and made substantial changes in the introduction. We would like to assign my article to the same reviewers.
Once again, thank you very much for your comments and suggestions. We appreciate for Editors/Reviewers’ warm work earnestly and hope that the correction will meet with approval.
Best regards,
Jing Yang.

Reviewer 2 Report
References formatting is inconsistent (e.g. lines 352-359).
There are some typos (e.g. Fig. 7).
Some important types of defects that are subjects of deep learning in practice detection are missing - e.g. typical defective (or missing) engraved serial characters.
And some methods that replace obsolete methods of space and dimensions inspections of large manufactured objects should also be mentioned e.g. laser scanning methods. Moreover, applications of advanced technical diagnostics like modal analysis or inspection of produced rotating parts that can be inspected at high-speeds only by the optical derotation method could also be mentioned.
The review paper should contain more general algorithms showing comprehensive principles and perspectives of data processing instead of extensive descriptive parts. Generally, trends and aims of development in the field should be formulated.
Author Response
Dear Editors and Reviewers,
Thank you for your letter and for the reviewers’ comments concerning our manuscript entitled “Using Deep learning to Detecting defects in manufacturing: A Comprehensive Survey and Current Challenges” (ID: materials-1006601). We are very excited to have this opportunity to revise our manuscript. Those constructive comments are valuable and very helpful for improving our paper and for guiding our future research. We want to extend our appreciation for taking the time and effort necessary to provide such insightful guidance. Mainly made the following changes。
- The paper was revised according to the Reviewer.Specific modification information cover letter.
- Polish the language;
- Deleted pictures that may cause copyright disputes;
- Added references 42-48,58-61,65-70,77,82-86,101,107,109,115-125,140,145-156;
- All revised parts in the paper have been marked in red.
- Added description of recurrent neural network.
We now detail our responses to each of the reviewers’ concerns and comments.
We tried our best to improve the manuscript and made substantial changes in the introduction. We would like to assign my article to the same reviewers.
Once again, thank you very much for your comments and suggestions. We appreciate for Editors/Reviewers’ warm work earnestly and hope that the correction will meet with approval.
Best regards,
Jing Yang.

Round 2
Reviewer 1 Report
The authors addressed all the reviewer's comments with sufficient attention and modified the paper consistently with the suggestions provided. The new version of the paper has improved significantly both in the presentation that is now much more accessible even by a reader not expert in the sector, and in the contents that now appear much more incisive.
In the new version, the cited techniques for identifying defects are adequately introduced and explained in detail. In addition, a detailed list of both traditional and innovative defect identification technologies is proposed. Now the methods based on deep learning are introduced and explained. This way the reader will be able to understand how these methods are used to identify defects. The authors also revised and modified the use of tables and figures, significantly reducing them. Now the tables and figures present are adequately commented. To avoid possible problems, the authors removed the unavailable with copyright.
47-51) Move this paragraph after the table 1.
54) “products.Surface” A space is missing between words
112) Introduce the numbered list
138) network[90].The - A space is missing between words
160) “Representative one-stage methods” Move after Table 3
162) “et al. Proposed” ? Maybe: et al. proposed
164) “Redmon et al. modified the network structure” I advise you to insert the citation number immediately after the name of the authors: Redmon et al. [96]. You could use this formatting throughout the paper.
217) Don't interrupt the flow of text with a Figure. Break text to next period.
237) Move the Table 4 after the bullet list of the research methods
278) “detection.It” A space is missing between words
294) “ndustries(Japan)[149].Recently” A space is missing between words
404-410) This looks like something you could have put at the end of the Introduction to prepare the reader. In this section you should show some conclusions: what does the study you have carried out lead to conclude, what are the future challenges?
Author Response
Dear Editors and Reviewers,
Thank you for your letter and for the reviewers’ comments concerning our manuscript entitled “Using Deep learning to Detecting defects in manufacturing: A Comprehensive Survey and Current Challenges” (ID: materials-1006601). We are very excited to have this opportunity to revise our manuscript. Those constructive comments are valuable and very helpful for improving our paper and for guiding our future research. Mainly made the following changes。
- The manuscript was revised according to the Reviewer;
- Rewrite the conclusion part;
- Revised some details of the manuscript;
- All revised parts have been marked red in the manuscript.
- See the attachment for details.
We tried our best to improve the manuscript and made substantial changes in the introduction. We would like to assign my article to the same reviewers.
Once again, thank you very much for your comments and suggestions. We appreciate for Editors/Reviewers’ warm work earnestly and hope that the correction will meet with approval.
Best regards,
Jing Yang.
